# Rhizobia Isolated from the Relict Legume *Vavilovia formosa* Represent a Genetically Specific Group within *Rhizobium leguminosarum* biovar *viciae*

**DOI:** 10.3390/genes10120991

**Published:** 2019-12-01

**Authors:** Anastasiia K. Kimeklis, Elizaveta R. Chirak, Irina G. Kuznetsova, Anna L. Sazanova, Vera I. Safronova, Andrey A. Belimov, Olga P. Onishchuk, Oksana N. Kurchak, Tatyana S. Aksenova, Alexander G. Pinaev, Evgeny E. Andronov, Nikolay A. Provorov

**Affiliations:** 1All-Russian Research Institute of Agricultural Microbiology, 196608 Saint Petersburg, Russia; chirak.elizaveta@gmail.com (E.R.C.); kuznetsova_rina@mail.ru (I.G.K.); anna_sazanova@mail.ru (A.L.S.); v.safronova@rambler.ru (V.I.S.); belimov@rambler.ru (A.A.B.); olony@ya.ru (O.P.O.); oxana-kurchak@yandex.ru (O.N.K.); tsaksenova@mail.ru (T.S.A.); ag.pinaev@gmail.com (A.G.P.); eeandr@gmail.com (E.E.A.); provorovnik@ya.ru (N.A.P.); 2Saint-Petersburg State University, 199034 Saint Petersburg, Russia; 3V.V. Dokuchaev Soil Science Institute of Russian Academy of Science, 119017 Moscow, Russia

**Keywords:** *Rhizobium leguminosarum* bv. *viciae*, *Vavilovia formosa* (Stev.) Fed., tribe Fabeae, evolution of symbiosis, housekeeping genes (*hkg*), symbiotic (*sym*) genes, group separation

## Abstract

Twenty-two rhizobia strains isolated from three distinct populations (North Ossetia, Dagestan, and Armenia) of a relict legume *Vavilovia formosa* were analysed to determine their position within *Rhizobium leguminosarum* biovar *viciae* (*Rlv*). These bacteria are described as symbionts of four plant genera *Pisum*, *Vicia*, *Lathyrus*, and *Lens* from the Fabeae tribe, of which Vavilovia is considered to be closest to its last common ancestor (LCA). In contrast to biovar *viciae*, bacteria from *Rhizobium leguminosarum* biovar *trifolii* (*Rlt*) inoculate plants from the Trifolieae tribe. Comparison of house-keeping (*hkg*: 16S rRNA, *gln*II, *glt*A, and *dna*K) and symbiotic (*sym*: *nod*A, *nod*C, *nod*D, and *nif*H) genes of the symbionts of *V. formosa* with those of other *Rlv* and *Rlt* strains reveals a significant group separation, which was most pronounced for *sym* genes. A remarkable feature of the strains isolated from *V. formosa* was the presence of the *nod*X gene, which was commonly found in *Rlv* strains isolated from Afghanistan pea genotypes. Tube testing of different strains on nine plant species, including all genera from the Fabeae tribe, demonstrated that the strains from *V. formosa* nodulated the same cross inoculation group as the other *Rlv* strains. Comparison of nucleotide similarity in *sym* genes suggested that their diversification within sym-biotypes of *Rlv* was elicited by host plants. Contrariwise, that of *hkg* genes could be caused by either local adaptation to soil niches or by genetic drift. Long-term ecological isolation, genetic separation, and the ancestral position of *V. formosa* suggested that symbionts of *V. formosa* could be responsible for preserving ancestral genotypes of the *Rlv* biovar.

## 1. Introduction

Root nodule bacteria (rhizobia) represent a useful model for studying the molecular and ecological mechanisms of evolution of symbiotic bacteria. Divergent evolution (intra-species radiation and formation of new species) by these bacteria is promoted by host plants, which elicit the selection pressures responsible for the genetic and ecological diversification of rhizobia [1,2,3]. This evolution may be traced using specialised symbiotic (*sym*) genes representing the accessory parts of bacterial genomes, which differ in their natural histories from housekeeping genes (*hkg*) representing the core parts of genomes [4]. As a result of co-evolutionary processes, symbiosis is formed between tightly co-adapted cross-inoculation groups of rhizobia and legumes, and their coevolution is directed by a set of symbiosis-specific genes from each partner [5,6,7]. In some rhizobia, *sym* genes are more susceptible to autonomous horizontal gene transfer than *hkg* genes, because they are located on plasmids—mobile elements of the genome [3]. This results in an intensive recombination of host specific and chromosomal markers [8]. For example, *Rhizobium leguminosarum* is composed of two biovars, which have diverged based on their plasmid-encoded host ranges [9]. Biovar *viciae* (*Rlv*) nodulates legumes from the Fabeae tribe, while biovar *trifolii* (*Rlt*) nodulates clovers from the Trifolieae tribe; however, they show a conservative chromosomal arrangement of *hkg* markers (Figure 1). 

Even so, divergent evolution of rhizobia is not restricted to *sym* genes. Application of the average nucleotide identity (ANI) method has demonstrated that a local *R. leguminosarum* population could be separated into five genomic species, differing in their *hkg* genes, representing their core genomes [10]. However, this speciation does not correlate with the diversification of *R. leguminosarum* into biovars *viciae* and *trifolii*, suggesting that *hkg* and *sym* gene evolution is controlled by different mechanisms. We decided to probe this hypothesis using the model of *Rlv*, a symbiont of the legume tribe Fabeae, which includes five genera: *Lens*, *Lathyrus*, *Pisum*, *Vicia*, and *Vavilovia* [11]. The latter genus consists of a single species, *Vavilovia formosa* (Stev.) Fed., a relict and endangered legume plant, which grows mostly in high-mountain regions of the Caucasus and Middle East [12,13,14,15]. Based on genetic and morphological markers, *Vavilovia* is closely related to *Pisum*, such that it is still sometimes attribute to this genus [16]. Based on a number of phenotypic traits, *V. formosa* is considered to be the closest living relative to the last common ancestor (LCA) of the Fabeae tribe [17]. Due to hard-to-reach niches, scarce populations, and the deeply growing roots of *V. formosa*, it is challenging to obtain a representative collection of its symbionts. Nevertheless, symbionts representing a local population in North Ossetia were previously described [18]. Most of the isolates were identified as *Rlv*, with some isolates belonging to the genera *Bosea*, *Tardiphaga*, and *Phyllobacterium*. A remarkable feature of these *Rlv* isolates was that the *nod*X gene was found in all strains [18]. This gene enables rhizobia to nodulate the highly selective Afghanistan pea lines, which have a specific allele of the *sym*^2^ gene, *sym*^2A^, which restricts nodulation by *Rlv* strains devoid of *nod*X [19].

We used the *R. leguminosarum*–*V. formosa* system to dissect microevolutionary processes within the cross-inoculation group formed by legumes of the Fabeae tribe. Specifically, we assessed a collection of *V. formosa* symbionts isolated from three geographically separated populations from North Ossetia, Dagestan (Russia), and Armenia. We compared nucleotide similarity of core *hkg* (16S rRNA, *dna*K, *glt*A and *gln*II) and accessory *sym* (*nod*A, *nod*C, *nod*D and *nif*H) genes of *V. formosa* strains with those of *R. leguminosarum* biovars *viciae* and *trifolii*. This approach enabled us to address the trade-off between the evolution of bacterial *hkg* and *sym* genes, which might be responsible for speciation and intra-species diversification processes, respectively.

## 2. Materials and Methods

### 2.1. Bacterial Collection and DNA Isolation

*V. formosa* plants with nodules were collected from three widely separated high-mountain populations in North Ossetia, Dagestan, and Armenia (Figure 2). They were sent to our laboratory by the Gorsky State Agrarian University in Vladikavkaz, Russia; the Mountain Botanical Garden in Makhachkala, Russia; and the Institute of Botany in Yerevan, Armenia. Sixteen plants were collected, each of which had 2–20 nodules on their roots. A total of 106 fast-growing rhizobia strains were isolated from nodules of *V. formosa* using a standard protocol [20]. All isolates were stored at −80 °C in an automated tube store (Liconic Instruments, Mauren, Lichtenstein) at the Russian Collection of Agricultural Microorganisms (RCAM, WDCM 966) in the All-Russia Research Institute for Agricultural Microbiology (ARRIAM) [21]. Information about these strains is available on-line in the database of RCAM (http://www.arriam.spb.ru). Isolates were cultivated at 28 °C and 220 rpm for 48 h in modified yeast mannitol broth (YMB) containing 1% sucrose [22]. DNA was obtained by the lysozyme–SDS–phenol–chloroform extraction protocol, with minor modifications [23]. The final concentration of DNA was measured on a SpectroStar Nano (BMG Labtech, Ortenberg, Germany), and preparations were diluted with water from Millipore Simplicity (Merck KGaA, Darmstadt, Germany) to a working concentration of 10 ng/µl. To optimise the number of samples, 1–3 strains were chosen at random from each plant for further analysis.

### 2.2. Polymerase Chain Reaction (PCR) Analysis

PCR with primers for *hkg* (16S rRNA, *dna*K, *glt*A, and *gln*II) and *sym* (*nod*A, *nod*C, *nod*D, *nod*X and *nif*H) genes were performed in a 30 µl reaction mixture, containing 10 ng template DNA, 10 pM of each primer, 1× buffer for Taq polymerase (Evrogen, Moscow, Russia), 4.5 nM of each dNTP (Helicon, Moscow, Russia) and one unit of Taq polymerase (Evrogen, Moscow, Russia). PCR reactions were performed on a T-100 thermal cycler (Bio-Rad, Hercules, CA, United States) with an initial denaturation at 95 °C for 3 min, 35 cycles of denaturation (30 s at 94 °C) and annealing (30 s at 50–62 °C), an extension of 1 min at 72 °C, and a final extension at 72 °C for 3 min. Primers and their annealing temperatures are listed in Table 1. PCR products were cleaned of residual enzyme and primers using a silica binding-based protocol [24]. All amplicons were directly sequenced on an ABI PRISM 3500xL Genetic Analyzer (Applied Biosystems, Waltham, MA, United States) at the Centre for Collective Use of Scientific Equipment’s “Genomic Technologies, Proteomics and Cell Biology” in ARRIAM. 

### 2.3. Sequence Analysis

Nucleotide sequences were processed using UGENE software (Unipro, Novosibirsk, Russia) [32], and deposited in GenBank under accession numbers listed in Appendix A. Reference sequences for the analysis of genes were taken from those *R. leguminosarum* genomes available in GenBank (Appendix A). Alignments were made using Molecular Evolutionary Genetics Analysis (MEGA) X software [33]. Further analysis was performed for separate genes, and also for *hkg* and *sym* gene concatenates. Nucleotide distances between and within the three populations were calculated in MEGA X using the *p*-distance method. These distances were also calculated within groups of all *V. formosa* (*Vaf*), vetch/pea (*Rlv*), and clover (*Rlt*) symbionts as reference groups. Statistical reliability of nucleotide distances was calculated using Statistica 12 [34]. In order to define the genetic and geographic factors responsible for nucleotide diversity, we used the Mantel test, which was performed using the *vegan* packet in RStudio [35]. To reconstruct phylogenies, neighbour-joining trees [36], with evolutionary distances computed by the maximum composite likelihood model and a bootstrap test (1000 replicates) [37], were constructed in MEGA X. Resulting dendrograms were visualised using the iTOL webtool [38]. 

### 2.4. Group Separation Statistics

Group separation analysis was performed for both individual genes and concatenates obtained for the *hkg* and *sym* gene groups. Measures of group separation of *Vaf* genes from those of *Rlv* and *Rlt* were calculated using the jackknife (JK) method, with average similarities, from BioNumerics (Applied Maths, Sint-Martens-Latem, Belgium) [39]. Each sequence from the pool was compared to all other sequences, and was assigned to the group (*Vaf*, *Rlv*, or *Rlt*) to which it was most similar. In cases of ambiguity, the tested sequence was assigned randomly to one of the groups. The depth of group separation was measured using the coefficient of nucleotide differentiation (N_st_) with the *p*-distance model and bootstrap test (1000 replicates) from MEGA X. This coefficient was calculated as (R_t_ − R_s_)/R_t_, where R_t_ is the nucleotide diversity (derived from *p*-distance) of both populations under comparison, and R_s_ is the nucleotide diversity within populations [40].

### 2.5. Sterile Tube Test Experiment

To determine the presence of nodulation of isolated rhizobia in symbiosis with different Fabeae plants, we performed sterile tube test experiments. Four *Vaf* strains obtained from different geographical regions were selected: *Rlv* Vaf-10, Vaf-12, Vaf-46 and Vaf-108. In addition, two strains were used as controls, *R. leguminosarum* bv. *viciae* 1079 (without the *nod*X gene) and *R. leguminosarum* bv. *viciae* A1 (with the *nod*X gene), with the latter being capable of effective symbiosis with Afghanistan lines of pea [41]. Nine legume plant species from the Fabeae tribe were used for estimating host specificity: *V. formosa*, *Vicia villosa*, *Vicia sativa*, *Pisum sativum* line SGE (European line), *Pisum sativum* (Afghanistan line), *Lathyrus pratensis*, *Lathyrus sylvestris*, *Lens culinaris* and *Lens nigricans*. *V. formosa* seeds were provided by the Gorsky State Agrarian University in Vladikavkaz; vetch, vetchlings, and lentils seeds—by the N. I. Vavilov All-Russian Institute of Plant Genetic Resources in St. Petersburg; and pea seeds—by the Laboratory of Genetics of Plant-Microbe Interactions of ARRIAM in St. Petersburg. Control rhizobia strains were provided by RCAM in ARRIAM.

Plant seeds were sterilised with concentrated H_2_SO_4_ and incubated at 4 °C until they germinated. They were then planted in one-litre glass cylinders containing vermiculite, with N-free liquid growth medium and a solution of microelements [42]. Seedlings were inoculated with 1 mL of a suspension of rhizobia containing approximately 10^7^ cells. As a negative control, 1 mL sterile water was added to the vessel. Plants were cultivated for 30 days in the growth chamber at 50% relative humidity, with a four-level illumination/temperature mode: night (dark, 18 °C, 8 h), morning (200 μmol m^−2^ s^−1^, 20 °C, 2 h), day (400 μmol m^−2^ s^−1^, 23 °C, 12 h), and evening (200 μmol m^−2^ s^−1^, 20 °C, 2 h). Illumination was provided by L 36W/77 Fluora lamps (Osram, Munich, Germany). Four replications were carried out for each sample. Results were recorded using a Carl Zeiss Stemi 508 stereo microscope with Zeiss Axiocam ERc 5S camera (Carl Zeiss Microscopy GmbH, Jena, Germany).

## 3. Results

A total of 22 fast-growing rhizobia strains were selected from nodules on *V. formosa* collected at three distinct locations: North Ossetia, Dagestan, and Armenia. Strains used and their geographic origins are listed in Appendix A. 

### 3.1. Population Diversity in Symbionts of V. formosa

A total of 198 sequences for nine genes of symbionts of *V. formosa* were acquired. Nucleotide similarity of each gene between populations did not differ statistically from that observed within populations (data not shown), so all three populations could be regarded as components of the same metapopulation of symbionts of *V. formosa*. 

The Mantel test demonstrated significant positive correlations between genetic and geographic distances for most of the genes, based on *p*-values (Table 2). The highest correlations were generally for *hkg*: *dna*K (0.34, *p* = 0.0004), *glt*A (0.45, *p* = 0.0001), *gln*II (0.41, *p* = 0.0001); and for the symbiotic gene: *nod*X (0.37, *p* = 0.0007). The rest of the *sym* genes also demonstrated positive correlations between genetic and geographic distances, though to a lesser degree, *nod*C (0.13, *p* = 0.039), *nod*D (0.26, *p* = 0.0037) and *nif*H (0.19, *p* = 0.0077). The 16S rRNA and *nod*A genes showed no statistically significant correlation between genetic and geographic distances. Although symbionts of *V. formosa* could be regarded as components of the same metapopulation, Mantel analysis shows the influence of geographical origin on gene diversity, which is more pronounced for *hkg* than *sym* genes.

### 3.2. Phylogenetic Analysis of Symbionts of V. formosa

Phylogenetic trees were constructed for individual genes (Appendix A) and for concatenates of *hkg* and *sym* gene groups (Figure 3 and Figure 4). Phylogenies of *hkg* genes showed no apparent differentiation of *Vaf* from the reference groups (*Rlv* and *Rlt*; Figure 3); however, there were some clades, particularly the Armenian and most of the Dagestan isolates, with some signs of separation. In contrast, phylogenies of the rhizobial *sym* genes demonstrated different patterns depending on the host plant. There was complete separation of *Rlt* with 100% bootstrap support, and less pronounced but still clear separation between *Rlv* and *Vaf* (Figure 4). At the population level, only the Armenian group displayed a trend to separation, while the Dagestan and North Ossetian groups were intermixed with each other. Strains carrying the *nod*X gene are widely represented on the dendrograms, but only the *Rlv* TOM strain, symbiont of Afghanistan pea and carrier of the *nod*X gene, is grouped with *Vaf* on the *nod*A, *nod*D, and concatenated *sym* phylogenies. We suppose, this grouping occurred due to their close place of origin, and not the presence of *nod*X, because the *Rlv* Vc2 strain, isolated from *V. cracca* in the United Kingdom, does not group with either *Vaf* or TOM. To conclude, *Vaf* isolates form separate a cluster with *Rlv* on the *sym*, but not *hkg* phylogenies.

### 3.3. Separation Statistics of symbionts of V. formosa from Rlv

The separation of *Vaf* genes from *Rlv* was measured by the JK method and by N_st_ coefficient. The first method indicated the “topology” of the separation, and the second method determined its depth. The results of these measures were consistent with one another (Figure 5). Two trends could be seen from these results: (1) differences in the degree of separation for different genes, which are statistically significant for *nod*C, *nod*D, and *nif*H genes in comparison with *hkg*, but was also evident when comparing genes within a category; and (2) a more pronounced separation of *Vaf* for concatenates than for individual genes. In particular, within symbiotic groups, the minimum separation was detected for *nod*A and the maximum for *nod*D (Figure 5).

Notably, JK measures were asymmetric for groups that were compared (Appendix A): *Vaf* strains were mostly similar to themselves, while *Rlv* strains were similar to both themselves and *Vaf*. This might be because the distances between some sequences in the *Rlv* group were bigger than the distance between the *Rlv* and *Vaf* groups.

### 3.4. Divergence of Hkg and Sym Genes in R. leguminosarum

Due to the design of our dataset, it would be incorrect to directly compare the nucleotide sequence similarities of genes between *Vaf* and *Rlv*/*Rlt*, since the *Vaf* group represented a naturally occurring metapopulation, while *Rlv* and *Rlt* were randomly chosen sets of genotypes. However, it was possible to compare the ratios of sequence similarity of *hkg*/*sym* genes within these groups. We demonstrated for the *Vaf* group that diversities of *hkg* and *sym* genes based on *p*-distance statistically did not differ, while for *Rlv*, the diversities of *sym* genes were statistically almost twice as high as those of *hkg* genes (Appendix A).

Previously, Kumar et al. [10] demonstrated that in the local *R. leguminosarum* population composed of bv. *viciae* and bv. *trifolii* strains, a pronounced diversity for *hkg* genes occurred, as determined using the ANI statistics. It resulted in the rhizobia being differentiated into five genomic (cryptic) species, which were not correlated to the host ranges. In order to look for a similar differentiation in our strain collection, we reorganised the group separation data (Appendix A), calculating the coefficients of average divergence (CAD) as 100% minus JK values. We demonstrated that a significant diversification of *sym* genes occurred only in the *Rlv* group, while for *hkg* genes, all three groups were highly diverse. For all three groups, the mean CAD values calculated for the *Vaf*, *Rlv*, and *Rlt* groups were significantly higher for *hkg* than for *sym* genes (Appendix A), which indicates that *hkg* are more ductile within biovars than *sym* genes.

### 3.5. Sterile Tube Test Experiment

Sterile tube tests were conducted with nine plant species, which represent all genera of the Fabeae tribe, including both cultured and wild-growing species. The goal was to estimate the significance of *sym* gene differences in *Vaf* strains on nodulation on different Fabeae plants. All four strains from *V. formosa* demonstrated a nod+ phenotype on all analysed plants (Figure 6). Therefore, *V. formosa* and its microsymbionts could be attributed to the cross-inoculation group of Fabeae–*Rlv*.

The colour and size of nodules varied in different plants; however, these features did not depend on the plant species. The effect of the rhizobial *nod*X gene on symbiosis was only seen for controls A1 (*nod*X+) and 1079 (*nod*X-).

## 4. Discussion

We analysed the genetic traits of rhizobia isolated from several populations of the relict legume *V. formosa*. Previously, we demonstrated most of these isolates might be attributed to *R. leguminosarum*, on the basis of 16S and internal transcribed spacer (ITS) sequences [43]. In addition, the *nod*C gene was sequenced, and it showed that isolated strains were closest to biovar *viciae* strains, which commonly nodulate plants from the Fabeae tribe. Another peculiarity was that all these strains contained *nod*X; this is typical of rhizobia nodulating *P. sativum* cv. *Afghanistan*, which grows in the Middle East and was known for its specific Nod factor receptor encoded by the *sym*2^A^ allele [44]. We also demonstrated that the *V. formosa* isolates differed from other bv. *viciae* strains in their *sym* gene sequences; however, they did not differ in their *hkg* sequences [45].

The aim of current study was to see whether these features were manifested in different populations, by analysing 22 *R. leguminosarum* strains isolated from *V. formosa* plants from three distinct populations of the Caucasus: North Ossetia, Armenia, and Dagestan. The list of analysed genes was extended to include *dna*K, *glt*A, and *gln*II for *hkg* genes, and *nod*A, *nod*D, *nod*X, and *nif*H for *sym* genes. Nucleotide sequences of these genes were used to differentiate between *V. formosa* isolates (*Vaf*) and reference *Rlv*/*Rlt* strains, using phylogeny and group separation methods. This allowed the divergent evolution within the *R. leguminosarum* species to be dissected, with a special emphasis on the trade-off between speciation and symbiotic diversification processes, as well as on the phylogenetic status of symbionts of *V. formosa* within bv. *viciae*.

### 4.1. Symbiotic Behavior of Isolates of V. formosa

The results of sterile tube tests suggest that *V. formosa* and its symbionts belong to the cross-inoculation group of the Fabeae *Rlv*. We did not detect differences between the inoculation of European and Afghan pea lines by symbionts of *V. formosa*, so the role of the *nod*X gene in *V. formosa* symbiosis remained unclear. For a more complete assessment of the symbiotic phenotypes formed by isolates of *V. formosa*, it would be necessary to conduct additional tests with a wider selection of strains and determine their nitrogen-fixing activity.

### 4.2. Divergent Evolution within R. leguminosarum

In order to dissect the divergent evolution of *R. leguminosarum* bv. *viciae*, we used two measures of group separation, JK and N_st_, which revealed both the separation and its depth; they also suggested that *Vaf* constitutes a genetically defined group within or very close to *Rlv*.

In order to analyse the trade-off between symbiotic diversification and speciation processes in rhizobia, we compared the divergence of *hkg* and *sym* genes. Previously, we demonstrated that the *Vaf* group diverged from the *Rlv* and *Rlt* groups for *sym* genes, but not for *hkg* genes [45]. In this report, with the help of group separation statistics, we demonstrated that intergroup divergence pertains to both gene categories (Figure 5), although in *hkg* genes it is not pronounced enough to result in the emergence of new species. The last statement is also supported by the ANI statistics for *Vaf* genomes compared to *Rlv*, as reported by Chirak [46].

For intra-group diversification, we demonstrated that different statistical approaches to compare *hkg* and *sym* gene diversification gave contrasting results. Specifically, *p*-distance analysis demonstrates (Appendix A) that diversification of *hkg* and *sym* genes was equal in the *Vaf* population, while in the *Rlv* and *Rlt* groups, *hkg* genes were less variable than *sym* genes. We also detected an almost twofold difference in the ratio of *hkg*/*sym* nucleotide polymorphisms between *Rlv* and *Vaf* rhizobia, which could indicate chromosomal diversification of strains, occurring together with the stabilising pressure of host plants on the *sym* genes of their symbionts (Appendix A).

Quite different results were obtained using the reorganised group separation (JK) data, which allowed us to demonstrate that within all three *R. leguminosarum* groups, divergence for *hkg* genes was much more pronounced than for *sym* genes (Appendix A). The same difference was demonstrated previously for *Neorhizobium galegae* biovars *orientalis* and *officinalis* [47]. Our data was consistent with that of Kumar et al. [10], who revealed a deep cryptic diversification for *hkg* genes in a local *R. leguminosarum* population, which did not correlate with the distribution of bacteria between the *viciae* and *trifolii* biovars.

Collectively, these data demonstrate that speciation and symbiotic diversification processes might represent independent vectors in rhizobia evolution, controlled by different population and molecular mechanisms operating in the core and accessory parts of bacterial genomes. Considering a pronounced positive correlation between nucleotide sequence similarity of some *hkg* genes and common geography (Table 2), these data suggested that *hkg* genes diverged under the influence of micro-evolutionary adaptations to local ecological factors, such as soil micro-niches, or from genetic drift.

It would be interesting to address the reasons for the different group separation values obtained for different *sym* genes. These differences might have a functional background, such as the transcription-activating impacts of NodD being induced by root-released flavonoids, which might differ in plants of the Fabeae tribe, resulting in maximal separation levels for *nod*D gene. In contrast to *nod*D, a minimal separation of *Vaf* and *Rlv* was detected for the *nod*A gene. This could result from similarities between chemical structures of the fatty acid tails in the Nod-factors of the rhizobia of *V. formosa* and those of other plants from this tribe. Therefore, *nod*A diversity might represent an interesting model for the analysis of the interplay between nucleotide diversity of *nod* genes and the chemical structure of the Nod-factor signal molecules.

### 4.3. Role of Geographic Factors in the Diversification of Rhizobia

The data on genetic differentiation of symbionts of *V. formosa* from the other *Rlv* strains in the absence of phenotypic differentiation for host specificity suggest that some non-host-dependent factors, such as geographic components, might be responsible for the gene diversification. The results of the Mantel tests (Table 2) suggest that a mechanistic topological (spatial) separation might represent no less a potent factor of rhizobia diversification than the selective pressures responsible for local adaptation. For the majority of *hkg* genes (*dna*K, *glt*A, and *gln*II), as well as for *nod*X, these correlations were highly significant (*p* < 0.001). For the majority of *sym* genes (*nod*C, *nod*D, and *nif*H), they were moderately significant (*p* = 0.0077–0.039), while for genes *16S rRNA* and *nod*A these correlations were absent (*p* > 0.05). The *sym* genes might have evolved mainly under selection pressures determined by host plants, which eliminated the correlations between the genetic and geographic diversities. However, for *hkg* genes, these pressures were generally low, and the diversity of *hkg* sequences was more dependent on geographic origin. A negligible correlation between topological and genetic diversity revealed for 16S rDNA might have resulted from strong stabilising selection monitoring the structure of this conservative gene.

### 4.4. Evolutionary Status of Symbionts of V. formosa

Based on the ancestral status of *V. formosa* and its long-term ecological isolation, we proposed that its rhizobia might also retain ancestral genetic and symbiotic traits. The uniform presence of the *nod*X gene in the *Vaf* group, unusual for rhizobia of the European pea lines, also supports this hypothesis.

Phylogenetic analysis demonstrated that *sym* gene diversity is clearly correlated with the separation of *R. leguminosarum* strains into biovars, while *hkg* gene diversity doesn’t show such a correlation (Figure 3 and Figure 4). However, the results of sterile tube tests suggested that *V. formosa* and its symbionts did not demonstrate host specificity within the cross-inoculation group of Fabeae–*Rlv*. These data suggested that during rhizobia microevolution, diversification of *sym* genes might precede phenotypic diversification for symbiotic traits. As such, the *V. formosa–Rlv* system might still be at the early stages of symbiotic evolutionary divergence, since the genetic differences were already pronounced, while phenotypic differences still had not occurred. Together with the proposed ancestral status of *V. formosa*, these data suggested that its strains might also preserve ancestral features, which were most clearly expressed at the level of concatenates of *hkg* and *sym* genes. Data indicating pronounced ancestral features for the *sym* gene arrangement within the bacterial genomes have been presented by Chirak et al. [46].

## 5. Conclusions

Genus *Vavilovia* is a phylogenetically compact group within the tribe Fabeae, which is sometimes included in its fellow genus *Pisum*. Nevertheless, *V. formosa* has some unique properties, such as its archaic phenotypic features and narrow habitat close to the centres of origin of cultivated plants. We analysed 22 *R. leguminosarum* strains isolated from three distinct populations of *V. formosa* collected in the Caucasus. Nucleotide sequence similarity of selected *hkg* and *sym* genes showed that statistically, all strains appear to compose one metapopulation, with a marked influence of geographic origin on *hkg* sequences.

We propose that that rhizobia strains isolated from *V. formosa* (*Vaf* group) represent a compact group within or very close to *Rlv* rhizobia, based on group separation statistics, including distances between and within groups, as well as JK asymmetry.

A comparison of different gene categories for nucleotide diversity suggests that while differences between sym-biotypes for *sym* genes in *R. leguminosarum* bv. *viciae* were elicited by host plants, the diversification of sym-biotypes for *hkg* genes was affected either by adaptation to soil niches or by genetic drift. Our data suggest that the speciation and micro-evolution of rhizobia might be controlled by different genetic mechanisms correlated to changes of core (*hkg*) and accessory (*sym*) genes, respectively.

## Figures and Tables

**Figure 1 genes-10-00991-f001:**
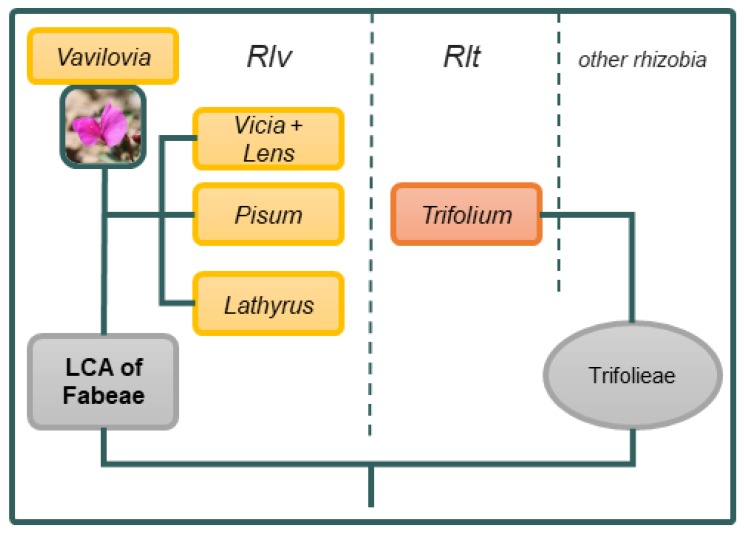
Plant hosts of *Rlv* and *Rlt* on the scheme of Fabeae evolution, based after Mikic [17]. Photo of *Vavilovia formosa* © Alexander Ivanov (North-Caucasus Federal University, Russia).

**Figure 2 genes-10-00991-f002:**
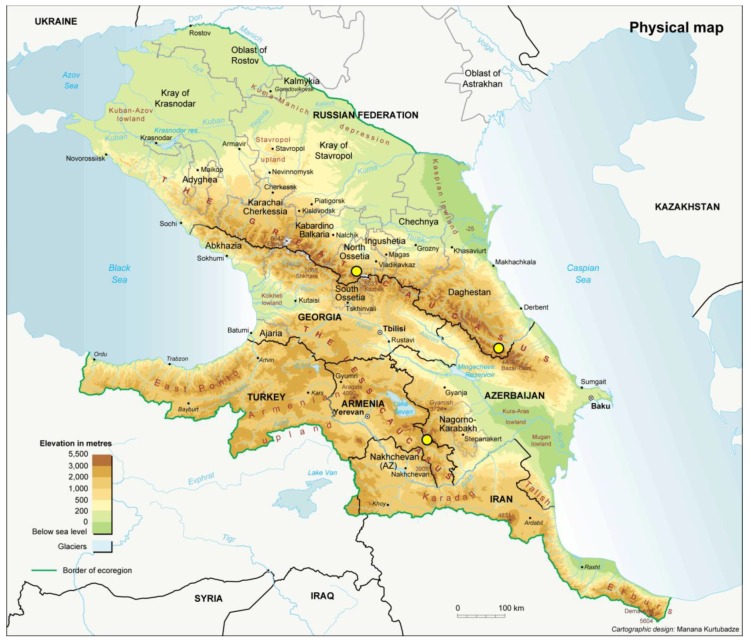
Origin of *V. formosa* populations from this study (black and white dots).

**Figure 3 genes-10-00991-f003:**
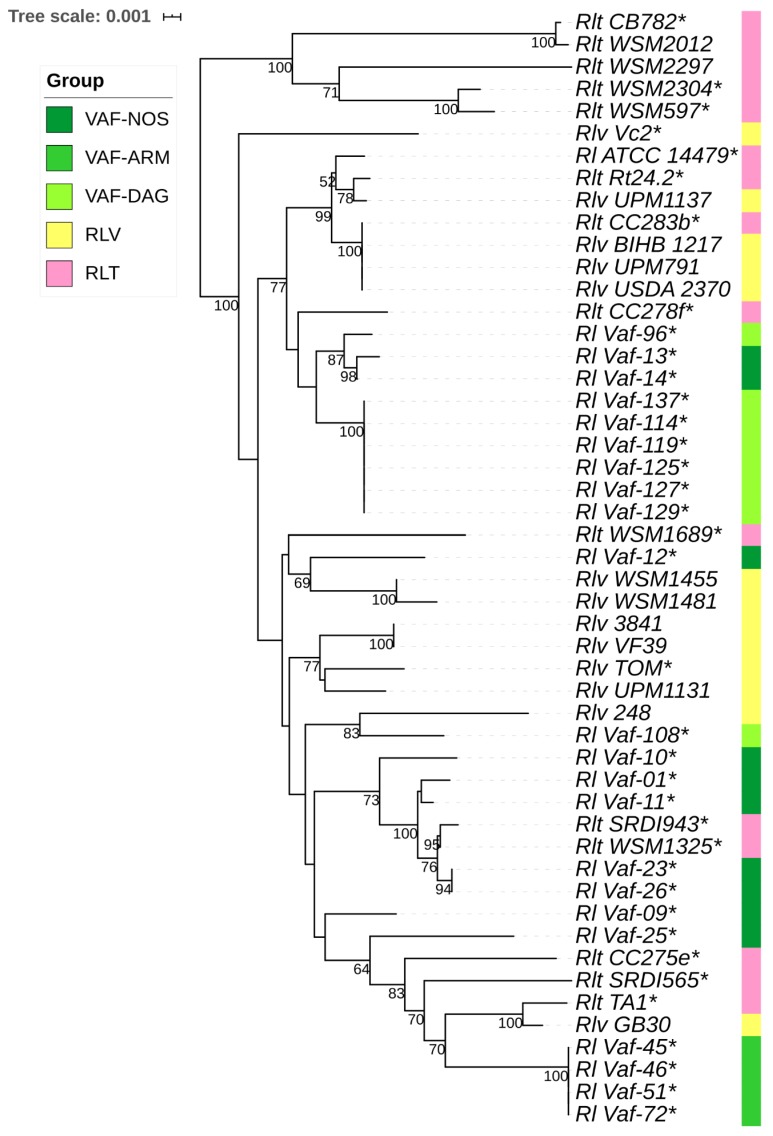
Neighbour-joining tree for concatenate of *hkg* (16S rRNA, *dnaK*, *glnA*, and *gsII*) of *Vaf* (light green—Dagestan, green—Armenia, dark green—North Ossetia), *Rlv* (yellow), and *Rlt* (pink) groups. The evolutionary distances were computed using the maximum composite likelihood method. Values of the bootstrap test (1000 replicates) exceeding 0.5 are shown next to the branches. * presence of *nod*X gene in a strain.

**Figure 4 genes-10-00991-f004:**
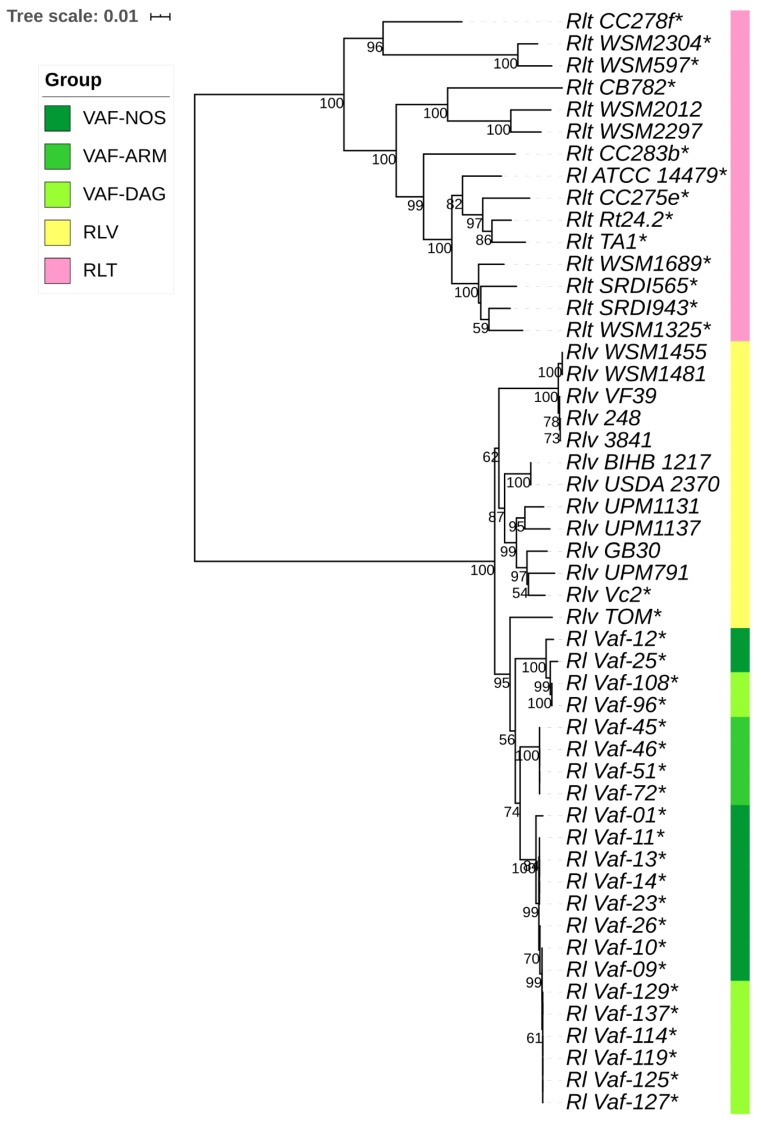
Neighbour-joining trees for concatenate of symbiotic genes (*nod*A, *nod*C, *nod*D, and *nif*H) of *Vaf* (light green—Dagestan, green—Armenia, dark green—North Ossetia), *Rlv* (yellow), and *Rlt* (pink) groups. The evolutionary distances were computed using the maximum composite likelihood method. Values of the bootstrap test (1000 replicates) exceeding 0.5 are shown next to the branches. * presence of *nod*X gene in a strain.

**Figure 5 genes-10-00991-f005:**
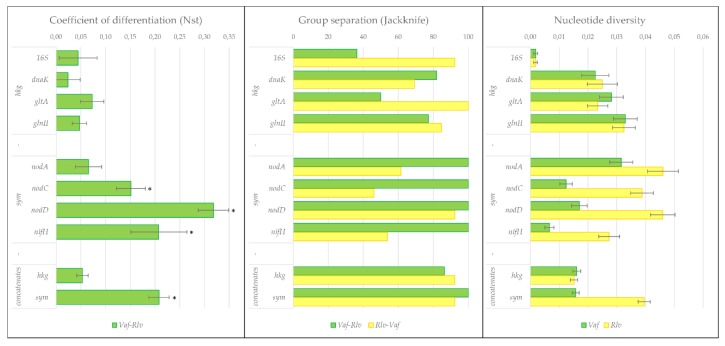
Group separation statistics. The panel represents summarised data for separation of *Vaf* from *Rlv*, including the coefficient of differentiation, group separation, and nucleotide diversity. Each measure is made for individual genes and concatenates of *hkg* and *sym*. In panel 1, values for *sym* that are significantly higher than those for *hkg* are marked with *. More detailed information can be found in Appendix A.

**Figure 6 genes-10-00991-f006:**
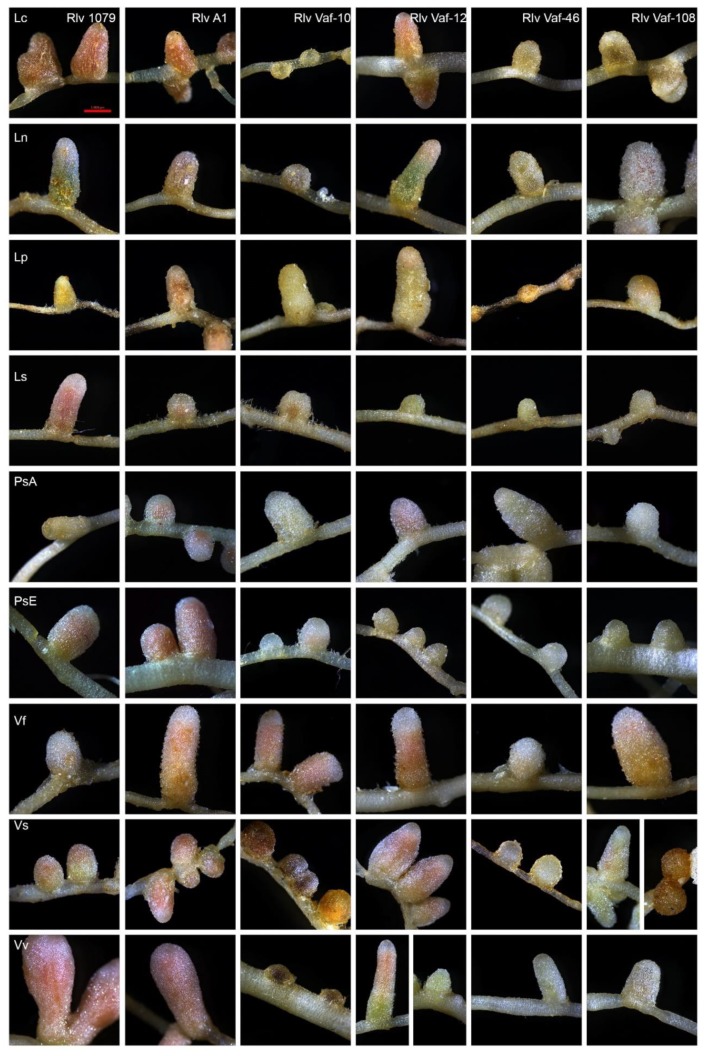
Results of the sterile tube tests. Pictures show nodule phenotype from each plant–strain combination. The scale for all images is 1 mm. Plants: (Lc) *Lens culinaris*, (Ln) *Lens nigricans*, (Lp) *Lathyrus pratensis*, (Ls) *Lathyrus sylvestris*, (PsA) *Pisum sativum* (Afghan line), (PsE) *P. sativum* SGE (European line), (Vf) *Vavilovia formosa*, (Vs) *Vicia sativa*, and (Vv) *Vicia villosa*. Rhizobia: *Rlv* 1079, *Rlv* A1, *Rlv Vaf*-10, *Rlv Vaf*-12, *Rlv Vaf*-46, and *Rlv Vaf*-108.

**Table 1 genes-10-00991-t001:** Primers used in this study.

Genes	Primers	Sequences	T °C	References
16S rDNA	27F	AGAGTTTGATCMTGGCTCAG	55	[25]
	1525R	AAGGAGGTGWTCCARCC		
*dna*K	dnaK1466F	AAGGARCANCAGATCCGCATCCA	62	[26]
	dnaK1777R	TASATSGCCTSRCCRAGCTTCAT		
*glt*A	gltA428F	CSGCCTTCTAYCAYGACTC	53	[27]
	gltA1111R	GGGAGCCSAKCGCCTTCAG		
*gln*II	GSII-1	AACGCAGATCAAGGAATTCG	55	[28]
	GSII-2	ATGCCCGAGCCGTTCCAGTC		
*nod*A	nodA-1	TGCRGTGGAARNTRNNCTGGGAAA	49	[29]
	nodA-2	GGNCCGTCRTCRAAWGTCARGTA		
*nod*C	nodCF	AYGTHGTYGAYGACGGTTC	57	[30]
	nodCI	CGYGACAGCCANTCKCTATTG		
*nod*D	NBA12	GGATSGCAATCATCTAYRGMRTGG	57	[31]
	NBF12′	GGATCRAAAGCATCCRCASTATGG		
*nod*X	oMP199	CCATGGGACCATCCAATGAAC	53	[19]
	oMP196	TTAAGCGACGGAAAGCCTTC		
*nif*H	nifHF	TACGGNAARGGSGGNATCGGCAA	62	[30]
	nifHI	AGCATGTCYTCSAGYTCNTCCA		

**Table 2 genes-10-00991-t002:** Pearson’s product-moment correlation between nucleotide and geographic distances for isolates of *V. formosa*, calculated by Mantel test in R studio (*vegan* packet).

Gene	Mantel Statistic r	*p*-Value
16S rRNA	0.0192	0.2978
*dna*K	**0.3379**	0.0004
*gln*II	**0.4051**	0.0001
*glt*A	**0.4494**	0.0001
*nod*A	0.0173	0.2807
*nod*C	**0.1266**	0.039
*nod*D	**0.2553**	0.0037
*nod*X	**0.3705**	0.0007
*nif*H	**0.1899**	0.0077

Statistically significant values are given in bold.

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
