# Peer review of "Rhizobia Isolated from the Relict Legume Vavilovia formosa Represent a Genetically Specific Group within Rhizobium leguminosarum biovar viciae"

_genes, 2019, doi:10.3390/genes10120991_

Round 1

Reviewer 1 Report

ABSTRACT:

General:

Sentences are generally too long. I’m having to re-read sentences to understand them. Sentences should be split up to avoid losing the point of the sentence. It’s difficult to keep track of the strains mentioned and how they relate to each other. Making clear, shorter sentences might make this clearer. As acronyms are being introduced, they are not consistently being used afterwards (i.e. Rhizobium leguminosa viciae is used interchangeably with its acronym, Rlv). As a reader who hasn’t yet established a strong association with the term and its abbreviation, I’m often having to backtrack to make sure I know what’s being referenced.

Specific:

Line 15-18: Consider rephrasing or separating into multiple sentences. Too much information and cuts for an opening sentence. Line 20-21: You mention R. leguminosarum biovar trifolii without introducing it along with Rlv. I would mention the two biovars of R. leguminosarum before talking about comparisons, especially as it’s an integral part of your research project. Line 27: Should be a comma after “plants” Line 28-31: Sentence too long. Concluding sentence should be clear. I would recommend moving the information about the LCA to a different location in the abstract.

INTRODUCTION:

General:

Good length, coherent sentences. Note the repeated use of conjunctive adverbs. Perhaps include information of biovars. I had to do research to get more context. Include a figure to better visualize the different genera of the Fabae tribe and how they relate to R. Leguminosarum, along with Rlv and R. leguminosarum biovar Trifolii (also include Rlt as an acronym?)

FIGURE 2

Good figure. Perhaps change the origin dots to a different colour to have a better contrast with the elevation colours. The dots don’t stand out, I had to search for them.

2.2 Polymerase chain reaction (PCR) analysis

The term “accessory” is now being used instead of hkg. Perhaps be more consistent with terms.

2.4 Group separation statistics

Line 129: “This analysis” is too general. Line 132: “In this method” not necessary.

2.5 Sterile tube test experiment

Line 141: “From the strains in this current study” This seems unnecessary. It’s well understood that you are referencing this study

RESULTS:

Population Diversity in symbionts of V. Formosa

What is the takeaway from a positive correlation between nucleotide and distances for rhizobia isolates? Concluding sentence needed to highlight the relevance of this information. The relevance of this information may be talked about later in the discussion, but provide a short conclusion here. What does this mean in regards to your main goals of the study set forth in the introduction? Line 175: “though to a lesser degree”. Is the lower significance important? Why mention this if it’s still significant. If the lesser degree is important, mention why.

TABLE 2

Strong title. Contrast to Table 1. Alignment of “Gene” to the left looks cleaner than if it was centered like some of the data in table 1. Be consistent with Table format between table 1 and 2.

Phylogenetic Analysis of symbionts of V. Formosa

Line 182-185: Perhaps this should be mentioned in the methods instead of just the results? Again, missing a concluding sentence or a sentence highlighting the relevance of this information. What does the lack of pattern with the NodX gene indicate?

FIGURE 4

You mention there was no pattern with the presence of the NodX gene. However, the star showing presence/no presence of the gene is a big focus in these figures. I suggest showing those particular results in a subtler way, given there’s no significance. With a quick look, I’m drawn to the stars and get the impression (Fig 4) that the presence of the NodX gene is relevant and significant. Good use of colour, very clear.

Separation statistics of symbionts of V. Formosa from Rlv

Please provide more information and topology and depth of group separation You talk about trends. Does this mean the results aren’t significant? This is not clear at all.

FIGURE 5

Indicate which results are significant. First box on the left is not aligned with the others Summarize the relevance of the significant data in the description I suggest including the Y axis title in all graphs.

Divergence of hkg and sym genes in R. leguminosarum

Note relevance of higher hkg

Steril tube test experiment

Repetitive from material and methods Good concluding sentence Perhaps add more information on nodulation in introduction

FIGURE 6

Add more detailed axes (names of plants and rhizobia) on addition to description

DISCUSSION and CONCLUSION:

Good recap. Line 260: “In this study” seems redundant. You use this line often throughout the manuscript. Alter subtitles to align more with the order of the results. There appears to be no adherence to the order in which the results were presented (i.e. the first topic in the discussion is the sterile tube test experiment, which was the last component presented in the results). Line 291-294: Good explanation. Would like to see this tied into the results. Line 316-324: Good discussion to address the different group separation values Overall, good discussion. Perhaps the text might be better organized if results and discussion were amalgamated. Strong conclusions

Reviewer 2 Report

The manuscript provides an expanded statistical analysis supporting the results of a previous study.  While the results and conclusions are not entirely novel, it does provide some new valuable information and insight on some evolutionary forces that may be responsible for the observations in the previous study. 

Reviewer 3 Report

This paper uses comparative genomics to explore the evolutionary relationship between twenty-two Rhizobium isolates from the relict legume Vavilovia formosa and an equivalent number of isolates derived from other members of the Fabeae tribe, but with a much wider geographical spread. 

From the nodulation tests described in Figure 6, it is argued that the Rl Vaf isolates belong to the same cross-inoculation group as other strains of Rl bv viciae. However, as indicated in Figure 4, there is one important feature which deserves more investigation. All the Rl vaf isolates analysed contain the supplementary genetic determinant nodX. This is thought to confer the ability to nodulate pea cultivar Afghanistan by conferring a second acetyl group to pentasaccharide lipochitin oligosaccharide Nod-factors. Unfortunately, the two strains of Rlv used for the inoculation tests in Fig. 6, namely 1079 and A1, are not used for the genomic study (Figs 3 and 4). In the limited amount of data presented, both strains appear to nodulate with V formosa, although 1079 (lacking nodX), appears to be less successful than Rlv A1 (which carries nodX). The data and text are unclear about whether the presence of nodX in strains of Rlv correlates with the ability to develop effective nodulation in V. Formosa host plants. NodX is uncommon in strains of Rl bv viciae, but it was found in most of the strains of Rl bv trifolii examined (Fig 4). This raises some important questions concerning the proposed status of V. formosa as the last common ancestor of the Fabeae tribe (as discussed in paragraph 4.4). I suggest that the authors conduct a few more tests on host specificity in order to answer the following questions: - (1) Are any or all of the strains of Rl bv trifolii that carry nodX capable of nodulating V. formosa and pea cv Afghanistan?; (2) Conversely, are strains of Rl Vaf (and also RL bv viciae TOM and A1) capable of nodulating any lines of Trifolium, particularly species isolated from the same geographical regions as the Rl Vaf isolates? On the basis of the evidence in Figs 3, 4 and 5 and the inoculation tests described in Fig 6, it would appear that the 22 isolates from nodules of V. Formosa are not sufficiently distinct to justify being grouped as a separate biovar of R. leguminosarum. Perhaps the authors could discuss this point and the fact that strain TOM (isolated in the same geographical region) is very similar to the Rl Vaf isolates (Line 194). In line 45 and through much of the research described in this paper, it is stated that the diversification of sym genes was correlated with the specificity of host plant species while the greater diversification of house-keeping genes may be driven by local adaptation to soil ecology. The authors should explain that this is probably the consequence of the fact that the sym gene clusters are carried on plasmids that are capable of horizontal transfer between indigenous soil-adapted strains of Rhizobium. Figure 2 is not cited in the text. What defines the limits of this ecological region? Line 23 should read “it will be necessary”.
